# HERON: A Novel Tool Enables Identification of Long, Weakly Enriched Genomic Domains in ChIP-seq Data

**DOI:** 10.3390/ijms22158123

**Published:** 2021-07-29

**Authors:** Anna Macioszek, Bartek Wilczynski

**Affiliations:** Faculty of Mathematics, Informatics and Mechanics, Univeristy of Warsaw, 00-927 Warszawa, Poland; a.macioszek@mimuw.edu.pl

**Keywords:** peak calling, ChIP-seq, histone methylation

## Abstract

The explosive development of next-generation sequencing-based technologies has allowed us to take an unprecedented look at many molecular signatures of the non-coding genome. In particular, the ChIP-seq (Chromatin ImmunoPrecipitation followed by sequencing) technique is now very commonly used to assess the proteins associated with different non-coding DNA regions genome-wide. While the analysis of such data related to transcription factor binding is relatively straightforward, many modified histone variants, such as H3K27me3, are very important for the process of gene regulation but are very difficult to interpret. We propose a novel method, called HERON (HiddEn MaRkov mOdel based peak calliNg), for genome-wide data analysis that is able to detect DNA regions enriched for a certain feature, even in difficult settings of weakly enriched long DNA domains. We demonstrate the performance of our method both on simulated and experimental data.

## 1. Introduction

Recent developments in next-generation sequencing (NGS) methods [1] have resulted in the emergence of many experimental protocols that allow us to examine various genome-scale phenomena, like epigenetic landscapes [2] or gene expression [3] on the level of detail that was previously unavailable. One of the NGS-based techniques that has been developed very early and has gained great popularity is the ChIP-seq protocol [2], which aims to localize protein binding regions in vivo, based on Chromatin Immuno-precipitation (ChIP). There are also other methods, such as ATAC-seq [4] or DNase-seq [5] experiments, that aim to identify regions of open chromatin and could be also a source for data for the algorithm we describe here; however, we will focus on the ChIP-seq data, as they are the most popular and our method gives the best results in the scenarios most likely to be seen in the ChIP-seq experiments.

In most NGS-based protocols, after the sequencing step, reads need to be mapped to the genome, so that the coverage of reads can be calculated for every position. Since we are interested in identifying regions of chromatin that are biologically relevant, the next step is usually to find the regions of enrichment, called peaks, in the coverage signal. In the case of ChIP-seq, these regions are expected to be the regions where the protein binds, while in the ATAC-Seq or DNAse-Seq, the regions of enrichment correspond to the open chromatin. The computational procedure of peak-calling has long traditions, going back to the vast methodology of signal processing [6]; however, it is not an easy task, and most of the methods available to researchers seem to have some disadvantages [7]. The problem of choosing the method to use is not made easier by the fact that there are literally dozens of these methods available, and it is very difficult to test them reliably [8].

The easiest, naïve approach is to arbitrary choose some threshold and consider every region with signal above it peaks and this approach has been used in some of the earliest ChIP-seq studies [2]. However, this approach fails to capture the complexity of the problem and is unable to reliably detect peaks in even slightly noisy data, which is a typical case in ChIP-seq.

Another approach is to assume the signal comes from a specific random distribution and identify regions where the signal is statistically significantly higher than expected, based on the model with parameters fitted on the data from the regions around it at some given level of significance. This approach is used by MACS [9], one of the most popular and widely used peakcallers. MACS assumes that read count data are distributed according to Poisson distribution. Other peakcallers can use Poisson assumption as well [10] or negative binomial [11], which has the advantage over the Poisson distribution that it can model data with variance higher than the mean. Other distributions are also sometimes used, such as the Gaussian in Sole-Search [12].

Another approach to enrichment analysis is to use Hidden Markov Models to represent the coverages, and it was used in the context of genomic peaks even before the NGS revolution [13]. Coverage of reads is considered to be a sequence of values emitted by states from some random distribution. The model is trained on the data using either Monte Carlo simulation, as in BayesPeak [14], or by Expectation Maximization algorithm, for HMMs known as Baum–Welch algorithm. Then the most likely sequence of states is found, usually with Viterbi algorithm [15]. Finally, the regions belonging to some specific state or subset of states are considered peaks. This is the approach we take in our method called HERON (HiddEn MaRkov mOdel based peak calliNg), schematically presented in Figure 1.

The character of the signal as well as the properties (length, frequency, enrichment value) of the enriched regions we want to identify depend greatly on the type of experiment and the way it was performed [16]. For example, ChIP-seq experiments against transcription factors tend to give short peaks with high enrichment over the background [9]. At the same time, ChIP-seq data detecting histone modifications, such as the repressive H3K27me3 or H3K9me3, tend to form very long (up to milions bp) peaks, often poorly enriched [10,17]. Various additional factors, like depth of sequencing, cell population heterogeneity, or quality of the antibodies used during the immunoprecipitation step in ChIP-seq experiment can also greatly influence the character of the signal, specifically the enrichment of the regions of interest and the level of noise around them [16].

Different peakcalling tools can be used to identify different kinds of peaks based on their type. We have compared several popular tools in terms of the features they provide to give the readers a guide in choosing the tool suitable for their application (see Table 1).

While there are many peakcallers that give satisfactory results for good-quality signals and manage to successfully identify narrow and highly enriched peaks, they usually struggle when discovering long, weakly enriched peaks, such as H3K27me3 domains. Here we show that approach based on Hidden Markov Models with continuous emissions can give acceptable results (sensitivity above 0.8 and 0.99) in such settings. We present results obtained on data from Roadmap Epigenomics project and on simulated data.

## 2. Results

### 2.1. Program Overview

The data in our approach are modeled by a three-state Hidden Markov Model. In the first step of the program, the analyzed genome is divided into adjacent windows of a certain width, which is 800 bp by default but can be changed by the user. Every window is considered to be in one of the three states: “no signal” (windows with zero or very little signal), “background” (or “noise”) or “peak” (or “enrichment”). Based on the read coverage in every window, the parameters of the model are estimated using Baum–Welch algorithm, and then the most likely sequence of states is determined using Viterbi algorithm. Coordinates of the windows in “peak” state (after merging adjacent ones) are considered coordinates of peaks.

The signal is assumed to come from either normal or negative binomial distribution. We found that using normal distribution often yields results with higher specificity and much faster EM algorithm convergence. Hence, this is the default behavior; however, the user can choose to use the negative binomial distribution if they think it suits their data better. One can provide multiple samples for a single peakcalling. In that case, two approaches are possible: either coverages in every window are treated as vectors instead of a single value and they come from a multidimensional random distribution, or they are simply summed.

For many ChIP-seq experiments, control samples (often called inputs) are also provided. They can be used to normalize the data and discover some universal biases present in both ChIP-seq and control signal. If the user provides a control file, we divide it into windows like the ChIP file. Then the value in every window in the analyzed data is set to $\log_{2}$(value_in_ChIP_sample / value_in_control_sample). To avoid dividing by zero or taking a logarithm of zero, first 1 is added to every window in both samples. Because this procedure usually does not produce integer values, the control file can be used only with Gaussian distribution (negative binomial distribution requires data to be non-negative integers).

### 2.2. Simulated Data

Results obtained on real data can never be perfectly assessed for their quality, as we do not have any outside knowledge where the peaks should be, so we can only rely on examining some downstream analysis, such as motif analysis (mostly in case of ChIP-seqs against transcription factors) or expression analysis (in the case of proteins that are expected to influence expression, such as H3K27me3 modification). This is why we decided to run tests on simulated data. In case of simulated data, we know where the peaks are, so we can precisely calculate the specificity and sensitivity of tested methods. Furthermore, simulating data allows us to assess in a controlled way how methods’ performance depends on the quailty of the data and width of sought peaks.

We generated various datasets with known peak coordinates. The datasets differed in three characteristics: (1) the width of the peaks used to generate them, (2) number of reads generated from peaks and hence coverage on peaks, and (3) number of reads generated as background and hence intensity of noise around the peaks. In particular, we used these datasets to assess how the choice of window width influences the results. We ran peakcalling on datasets with various peak width, using various window widths, and compared obtained peak sets to the peak sets used to generate the dataset (i.e., the peaks we want to discover). We used the Jaccard index as a measurement of quality of the results. In Figure 2, we present how the Jaccard index depends on used window width and the width of the sought peaks. One can see that our approach gives reasonable results mostly for longer peaks (>3000 bp). As far as the choice of window width is concerned, it does influence the results to some extent; however, the result quality is quite robust to the choice of this parameter. Based on the outcomes of our simulations, we recommend using width between 300 and 1000, which seems to yield very good results for all peak sizes; however, the user may choose a specific window size that fits a particular specific application. Using shorter window width theoretically allows one to detect peaks more precisely; in particular, using width equal to one basepair would allow us to discover peaks of any length (because the final peaks will be built from the windows, so their width will be a multiple of window’s width). In practice, however, using short windows causes difficulties with the algorithm’s convergence and results in calling many artifacts, as it requires an estimation of many more transitions than with longer windows while being exposed to a much higher noise ratio. Peak resolution equal to 1000 is usually enough when one is interested in discovering long domains, and our method is mainly designed to discover such domains.

We used simulated data with various quality (i.e., various noise and enrichment ratios) and various peak widths to assess our method and compare it to other peakcallers. We used MACS2 [9], because of its popularity, SICER [18], which is also designed to identify long domains, and BayesPeak [14], another peakcaller that uses Hidden Markov Models. We ran MACS2 with two settings—one with default options and one with additional option “broad” that makes it search for longer peaks. Our peakcaller was tested with two settings—with Gaussian distribution and negative binomial distribution. We did not simulate controls, so we did not test it by normalizing on the control sample.

In Figure 3, we present our results for various simulated datasets that differ in the width of the peaks used to simulate the data and the average enrichment on peaks. For every method of peakcalling, we compared obtained peaks with the set of “real” peaks, i.e., peaks used to simulate the data. We used three different measures to assess the results: Jaccard index (Figure 3a), True Positive Rate (Figure 3b), and False Discovery Rate (Figure 3c).

We discovered that our approach gives better results (in terms of Jaccard index) than MACS and BayesPeak for long peaks (>3000 bp). SICER can give better results than HERON for such peaks, but not when the enrichment is small (average coverage simulated on peaks lower than 5). Additionally, when the enrichment is small, HERON tends to outperform MACS and BayesPeak, even for shorter peaks. We noticed that MACS usually has very good specificity, often at the cost of low sensitivity; it always calls very few artifacts, but at the same time very few true positives; for long peaks, MACS sometimes did not call any peaks at all, even with the “broad” option. On the contrary, our tool tends to call more false positives than MACS, but also more true positives. Furthermore, peaks called by our method usually resemble the target peaks as far as their width is concerned better than MACS; MACS tends to call short peaks, even with the “broad” option. Overall, the Jaccard index of our results is consistently better than that of MACS’s and BayesPeak’s results when peaks are long or enrichment is poor; additionally, the index is better than that of SICER’s results when both these conditions are met. For short peaks, especially with very high enrichment, MACS tends to give better results (Appendix A).

### 2.3. H3K27me3 from Roadmap Epigenomics

We tested our program on publicly available data from Roadmap Epigenomics project [19], which aims to create a large database of epigenomics features in human genome. It collects data from various NGS-based experiments (mostly ChIP-seqs for histone modifications) from various human tissues. Apart from the raw data, results from some downstream analyses are available too, in the case of ChIP-seq experiments, sets of peaks called using MACS. We decided to test our program on ChIP-seq data on H3K27me3.

We ran our program in three different settings: (1) with negative binomial distribution, (2) with Gaussian distrubution, and (3) with Gaussian distribution and using an input file as a control sample. We compared our results to the peaks available in Roadmap Epigenomics, i.e., peaks called with MACS, using an input file, and to the peaks called by SICER. We ran the analyses on 10 example samples from six tissues.

In Figure 4, we present the results, averaged over tissues. We observed that our program tends to call much longer peaks—mean peak length is around 23K bp for Gaussian distribution, 29K for Gaussian with control sample, and around 48K bp for negative binomial one, while for MACS and SICER peaks it is only 1400 and 4400, respectively (Figure 4c). Considering the fact that H3K27me3 usually forms long domains, overlapping many genes and spanning thousands of nucleotides, it seems that our method gives results with higher sensitivity than SICER and with higher specificity than MACS.

H3K27me3 is placed on the genome by the Polycomb proteins, and it is considered a repressive mark; hence, one might expect genes that locality inside H3K27me3 domains to have lowered expression. We the checked expression of genes that overlap with peaks called by all the tested methods and compared it with the expression of all genes (Figure 4e). We observed that genes within H3K27me3 peaks tend to have lower expression, as expected; furthermore, genes within peaks called by our methods have lower expression compared with genes within peaks published in Roadmap Epigenomics, again suggesting that our method might be better suited than MACS for calling long peaks like H3K27me3 domains.

### 2.4. Multiple Samples

One can use multiple samples in a single peakcalling, for example, multiple technical replicates of the same experiment. This could be especially helpful in the case of poorly enriched and noisy data. For such data, it is easy for any peakcaller to miss some weakly enriched peaks or mistake accidental peaks emerging from background for actual peaks; hence lowering both sensitivity and specificity of the method. However, when we use multiple samples, we can take advantage of the fact that the actual peaks—including the weak ones—will be highly repetitive between samples, while the artifacts’ localization will be mostly random and should not correlate with other samples. Therefore, using multiple samples helps to discover weakly enriched peaks and distinguish them from the artifacts, improving sensitivity and specificity.

In Figure 5, we show how the number of files used in peakcalling improves the results. We tested it on both real data from Roadmap Epigenomics and simulated data. In Figure 5a, we show how Jaccard index increases with increasing number of files provided for simulated data. It seems that especially for shorter peaks (<5K), using additional samples can substantially improve the results. In Figure 5b, we compare two approaches to using multiple files: (1) files are merged together, and the signal in every window is a sum of coverages in this window in all the files; (2) files are treated separately, and the signal in every window is a vector of coverages in this window in all the files. It seems that, especially for long peaks, the difference between the two approaches is negligible. In Figure 5c, we show results for Roadmap Epigenomics data: we show that the expression of genes overlapping with peaks called on all the files available for the given tissue is lower than expression of genes overlapping with peaks called using only one file, suggesting that peakcalling on all the files gives more reliable results. However, the length of the peaks called on all files is shorter than for separate files. It could mean that using all the files produces more conservative consensus set of peaks; the regions called with this method are more reliable at the cost of their length.

## 3. Conclusions

In this paper, we have studied the performance of a novel HMM-based peakcaller for ChIP-seq data in various contexts of real and simulated datasets with varying data characteristics (peak length, coverage, enrichment, noise). We compared it to other approaches: MACS—the most popular tool based on enrichment hypotheses testing; SICER—a peakcaller intended to work with signal producing long domains; and BayesPeak—arguably the most popular HMM based peak caller.

Using our simulated ChIP-seq data we were able to show, perhaps not surprisingly, that in the case of narrow peaks with high enrichment, the peakcalling ability of MACS or SICER is indeed close to perfect and should be recommended for use in such scenarios. However, in situations where the enrichment values are lower (especially for signal enrichments below 5 times over the background level) and spread over longer domains (especially ≥ 20 kbp), we see that the available methods do not perform as well anymore. In a wide range of these scenarios (50 kbp ≥ peak length ≥ 5 kbp, enrichments ≥ 1.5), our HMM-based peakcaller was able to perform much better than the state-of-the-art methods. This high-level performance was obtained under a wide range of method parameters (e.g., 300 bp ≤ window width ≤ 5000 bp). When we focused on these types of synthetic data (see Figure 2), we could clearly see that our method not only outperformed the other methods in the overall measure of Jaccard index, but it showed a vast improvements in sensitivity for even the lowest enrichments, in cases where the peaks were longer than 5 kbp, while maintaining modest increase over MACS and BayesPeak in the most difficult case of shorter, weakly enriched peaks. All of this was achieved without detrimental false positive rates, especially in the case of data with longer enrichment regions. The only tool with comparable performance seems to be SICER [17]; however, while it shows increased specificity over HERON, its sensitivity to longer peaks remains significantly lower than that of HERON.

Since these scenarios are relevant for ChIP-seq data analysis of broadly deposited histone marks, such as H3K27me3, we have also tested our method on experimental data from such ChIP-seq experiments from Roadmap Epigenomics project [19]. By comparing gene expression measured in the same cell populations, we could compare the quality of peaks detected by different methods by assuming that the lower the expression of the genes detected to contain the H3K27me3 mark, the better the peak calling procedure. In most cases, both HERON and MACS detected peaks in genes showing significantly lower gene expression. However, our results (Figure 4), and especially the ones obtained with the Gaussian distribution, showed much lower expression levels, while at the same time, we detected much fewer peaks with lengths significantly higher than those detected by MACS.

Lastly, we have tested the potential of our method to detect peaks in situations where multiple data files were available in an experiment. It is an important feature for the analysis of ChIP-seq data, as they can frequently display significant variance between replicates. We can show that our method can strongly benefit from the presence of additional replicates in the case of shorter peaks, while in the case of longer peaks, the performance increase seems to be moderate.

Overall, we are confident that our new HMM-based method for peak detection will be a useful tool for researchers studying chromatin modifications with long enrichment domains, such as histone modifications. We have made our tool freely available to the public at https://github.com/maciosz/HERON (accessed on 24 July 2021), and, given the results we present here, we expect it to become popular among scientists working on epigenomics.

## 4. Methods

### 4.1. Program Details

Program is based on hmmlearn (version 0.2.2) [20] package for python; in particular, training HMM and finding most likely sequence of states is performed by hmmlearn. The training is performed with Baum–Welch algorithm, and the most likely sequence of states if found with Viterbi algorithm.

When input files are bams, not bedgraphs, coverage in windows is calculated using pysam [21] package for python. For Gaussian distribution, mean coverage is calculated for every window, and for negative binomial distribution, summaric coverage.

Algorithm initializes parameters as follows: means of distributions of emissions are set to 0, 0.5, and 0.99 quantile of data. The user can change those values or provide their own initial means. The covariance matrix is initialized as diagonal sample covariance matrix for each state. The user can change it to be full, spherical (each state has a single variance value) or tied (all states has the same full covariance matrix). If negative binomial distribution is used, parameters *p* and *r* are calculated as follows:$$p=\frac{mean}{var}$$
$$r=\frac{mean^{2}}{var-mean}$$

Scores are assigned to each called peak. For every window that belongs to the peak, we know the coverage of reads in it and we calculate posterior probability that this window belongs to the state “peak”. If we assume that peak *i* consists of $x_{i}$ windows, then two $x_{i}$-element sets of values can be defined for it: set of coverages and set of posterior probabilities. Four types of scores are then calculated:Mean coverageMaximum coveragePosterior probability that these windows are in state “peaks”, i.e., product of the posterior probabilities in the individual windowsMaximum from posterior probabilities.

All the scores are saved to the [output_prefix]_peaks.tab file. By default, mean coverage is saved to [output_prefix]_peaks.bed (bed format supports only one column with score). The user can change this behavior with the “–score” parameter.

### 4.2. M-Step for Negative Binomial Distribution

When negative binomial distribution is used during the EM algorithm, parameters *p* and *r* are updated in turns, i.e., in every *i*-th iteration, *r* is updated, and in every (i+1)-th iteration *p* is updated. The maximum likelihood estimator for *p* is calculated as follows:$$p_{j}=\frac{\sum_{t}P_{j,t}*r_{j}}{\sum_{t}P_{j,t}\left(x_{t}+r_{j}\right)}$$
where $P_{j,t}$ means posterior probability that window *t* is in the state *j*; $r_{j}$ is value of parameter *r* for state *j*; and $x_{t}$ is emitted value in window *t*.

Maximum likelihood estimator for *r* parameter is found iteratively. We begin from the current estimation of *r*, obtained in the previous EM iteration. The derivative of log likelihood function by *r* is calculated at this point as follows:$$\frac{dl}{dr_{i}}=\sum_{t}P_{i,t}*\left[\Psi \left(x_{t}+r_{i}\right)-\Psi \left(r_{i}\right)+ln\left(p_{i}\right)\right]$$
where Ψ denotes the digamma function.

In the (i+1)-th iteration, $r_{i+1}$ is set to $r_{i}+\Delta_{i+1}$. $\Delta_{i+1}$ is calculated as follows:If derivative in $r_{i}>0$:(a)If $\Delta_{i}>0$: $\Delta_{i+1}=\Delta_{i}$(b)If $\Delta_{i}<0$: $\Delta_{i+1}=-\frac{1}{2}\Delta_{i}$(c)If $\Delta_{i}=0$ (i.e., i=0): $\Delta_{i+1}=r_{0}$If derivative in $r_{i}<0$:(a)If $\Delta_{i}>0$: $\Delta_{i+1}=-\frac{1}{2}\Delta_{i}$(b)If $\Delta_{i}<0$: $\Delta_{i+1}=\Delta_{i}$(c)If $\Delta_{i}=0$: (i.e., i=0) $\Delta_{i+1}=-\frac{1}{2}r_{0}$

The iterations continue until derivative in current *r* estimation is lower than some threshold (currently set to $1\times 10^{-5}$).

### 4.3. Peakcalling

Peakcalling with our method was done using default parameters for simulated data; in particular, in Figure 2, Gaussian distribution was used. For H3K27me3 peakcalling, the resolution was set to 1000 instead of the default 800. Peakcalling with MACS2 [9] was performed using version 2.2.6, either with default parameters or—if specifically stated so in text—with the “–broad” option. Peakcalling with BayesPeak [14] was done using version 1.30.0 with default options. Peakcalling with SICER [18] was done using version SICER2 1.0.3 with default options.

### 4.4. Roadmap Epigenomics Data

Three types of data were downloaded from Roadmap Epigenomics project (accessed on 24 July 2021): aligned reads, called peaks and expression table. Reads were downloaded in tagAlign format from https://egg2.wustl.edu/roadmap/data/byFileType/alignments/unconsolidated/H3K27me3/ and https://egg2.wustl.edu/roadmap/data/byFileType/alignments/unconsolidated/Input/.

The tagAlign files were transformed into bed format and then to bam format using “bedtools bedtobam” (version 2.27.1) [22]. Peaks were downloaded in broadPeak format from https://egg2.wustl.edu/roadmap/data/byFileType/peaks/consolidated/broadPeak/ (accessed on 24 July 2021) and transformed to bed format for downstream analysis. Expression table in RPKM for protein coding genes was downloaded from https://egg2.wustl.edu/roadmap/data/byDataType/rna/expression/ (accessed on 24 July 2021).

We ran analyses for 10 samples: from adult liver (3 samples), fetal brain, brain hippocampus middle (3 samples), spleen, thymus, and brain germinal matrix.

### 4.5. Results Analysis

Jaccard index, True Positive Rate, and False Discovery Rate were calculated using bedtools [22]. For Jaccard index, “bedtools jaccard” tool was used. For TPR, number of nucleotides considered True Positives were calculated with “bedtools intersect” as a total number of nucleotides common between the sets of real and called peaks. To obtain TPR, the number was divided by the total number of nucleotides in real peaks (i.e., summaric length of real peaks). For FDR, number of nucleotides considered False Positives were calculated with “bedtools subtract” as a total number of nucleotides of intervals present in the called peaks, but absent in the real peaks. To obtain FDR, the number was divided by the total number of nucleotides in called peaks.

To assess expression of different subsets of genes, we found genes overlapping with peaks (overlap by 1 nucleotide was sufficient) using “bedtools intersect”. We used gene annotation published in Roadmap Epigenomics project (accessed on 24 July 2021): https://egg2.wustl.edu/roadmap/data/byDataType/rna/expression/.

Plots were made using ggplot2 (version 3.3.5) [23] pacakge from R.

Screenshot from genome browser was made with Integrated Genome Browser (version 9.1.8) [24].

### 4.6. Simulated Data

All the simulated datasets are for chromosome 21 of human genome (version hg38). First, we generated the desired peak coordinates; as a template, we used coordinates of enhancers in fetal brain from EnhancerAtlas project [25]. The coordinates were transfered from hg19 to hg38 with liftOver tool from UCSC [26]. On chromosome 21, 427 enhancers were succesfully transferred. To generate peak sets with fixed peak width, for every peak, we kept the beginning and set the end as (beginning + desired length). This way we obtained 18 sets of peaks; in each all the peaks have the same width. We generated sets of peaks with length 50, 100, 200, 400, 600, 800, 1000, 1500, 2000, 2500, 3000, 5000, 7500, 10,000, 20,000, 30,000, 40,000, and 50,000. For the longest widths many peaks started overlapping, so we removed them; to keep the same number of peaks in every set, we also removed these peaks in the sets where they did not overlap. At the end, we obtained 18 sets, each with 157 peaks.

In the second step, we generated reads from these peaks using ChIP-sim package (version 1.3.1) [27] from Bioconductor (R). We used scripts provided with the package’s vignette, with small changes allowing for using our own peak coordinates instead of simulating them; the actual scripts we used are available at https://github.com/maciosz/NGS_simulation (accessed on 24 July 2021). For every dataset, we generated 1,000,000 reads, apart from the datasets with longest peaks (20,000 and longer), for which we generated 5,000,000 reads. Additionally, we generated 10,000,000 reads uniformly sampled from the whole chr21.

In the third step, the simulated reads were mapped to chr21 using Bowtie2 (version 2.1.0) [28] with default parameters. Here we obtained one bam file with reads uniformly sampled from the whole chromosome, which will be used to simulate noise (background) and 18 bam files with reads sampled from peaks.

In the fourth step, we generated final datasets by sampling the mapped reads with “bedtools sample”: each dataset is a mix of reads sampled from noise bam and one of the peak bams. The number of reads to sample was determined by the desired coverage: to obtain average coverage for noise equal to x, we sample x * genome_size / read_length reads from noise bam, and to obtain average coverage for peaks equal to y, we sample y * peak_width * peak_number / read_length reads from the bam with peaks of peak_width width. Note that it means that after combining both sets of reads, the average coverage on peak regions will be actually equal to x + y. For every peak width, we generated datasets with average coverage for peaks equal to 2, 3, 5, 10, 15, 20, or 25 and average coverage for noise equal to 0.25, 0.5, 1, 2, 3, or 5; that way, for each of the 18 peak widths, we obtained 7×6 = 42 variants of enrichment and noisiness.

The simulated data presented in Figure 2 were generated with average coverage from peaks equal to 5, and from noise equal to 1; in Figure 5, average coverage from peaks is equal to 3, and that from noise is 3. In Figure 3, and Appendix A average coverage from noise is equal to 3.

The bam files were sorted and indexed using samtools (version 1.3.1) [29]; the coverage tracks were generated from bam files using bedtools and converted to bigwig format using bedGraphToBigWig [30] tool from UCSC.

## Figures and Tables

**Figure 1 ijms-22-08123-f001:**
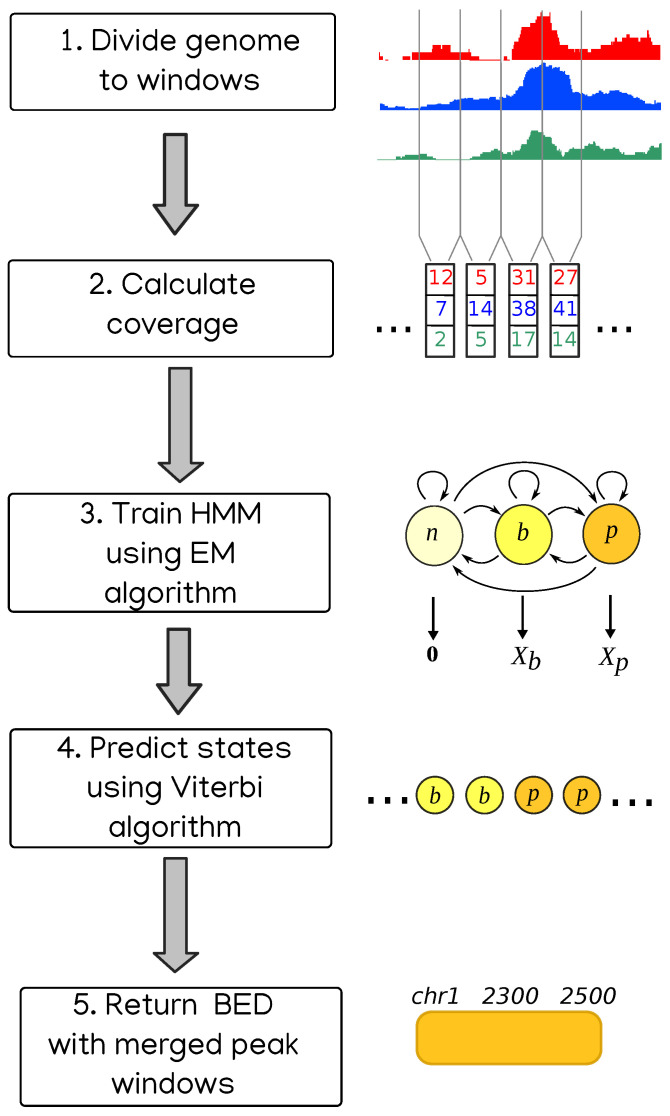
The schematic depicting consecutive stages of the HERON peak calling workflow.

**Figure 2 ijms-22-08123-f002:**
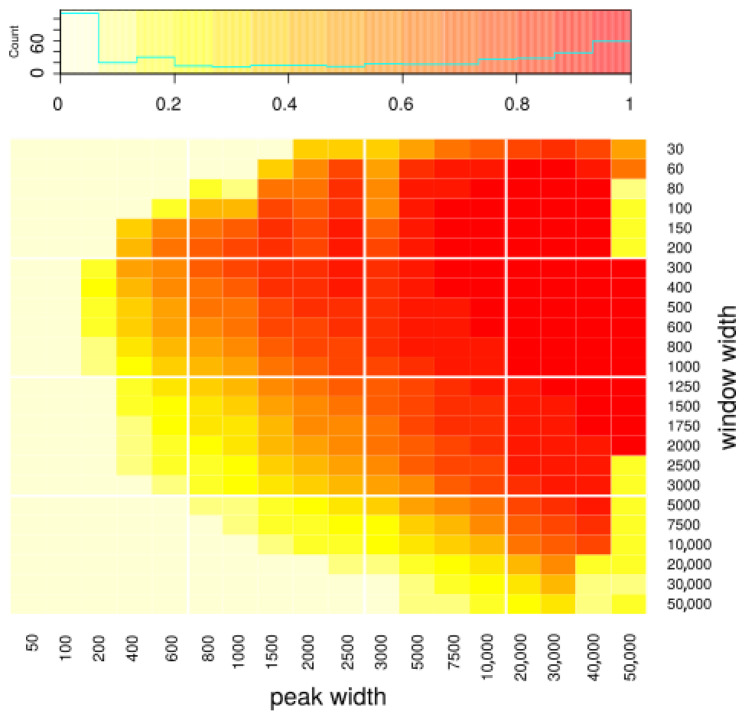
Assessment of peakcalling on simulated data depending on window width and the length of the peaks used to simulate the data, i.e., the peaks that we want to discover. Color represents Jaccard index, defined as |predicted * real| / |predicted + real|, where |predicted|—summaric size of regions predicted as peaks; |real|—summaric size of real peaks; i.e., simulated ones that we wanted to discover; A*B—intersection; A+B—union. The *x* axis represents the peak width, and the *y* axis represents the window width used during the peakcalling.

**Figure 3 ijms-22-08123-f003:**
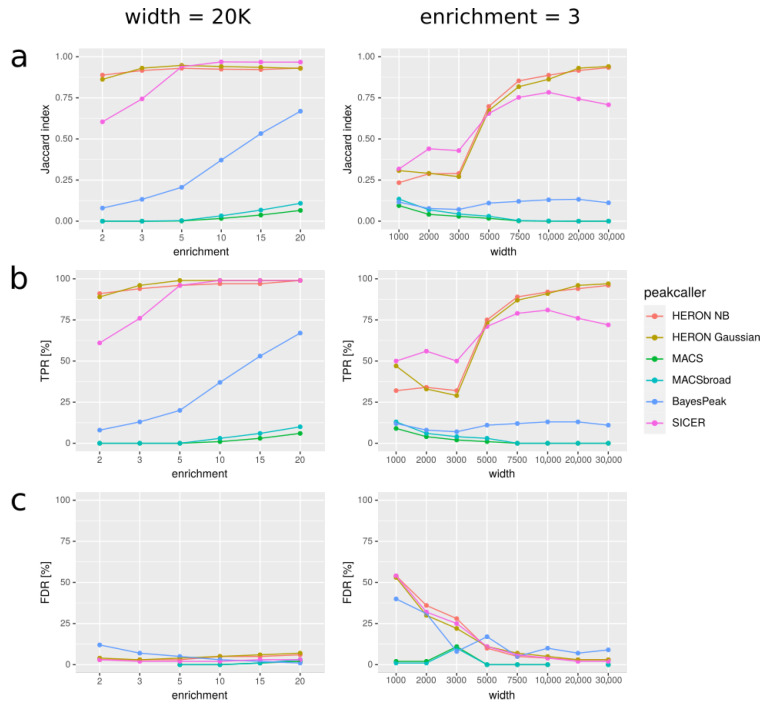
Comparison of peakcallers on simulated data for two different settings. In the first column, all the simulated peaks are 20K bp long, and simulated average enrichment varies. In the second column, average enrichment on peaks is constant and equal 3, and the peak length varies. In the rows, three measures are shown: (**a**) Jaccard index: |predicted * real| / |predicted + real|, (**b**) TPR: |predicted * real| / |real|, and (**c**) FDR: |predicted ∖ real| / |predicted|; where: |predicted|—summaric size of regions predicted as peaks; |real|—summaric size of real peaks, i.e., simulated ones that we wanted to discover; A*B—intersection; A+B—union; A ∖ B—difference (A and not B). “HERON NB” means HERON with negative binomial distribution.

**Figure 4 ijms-22-08123-f004:**
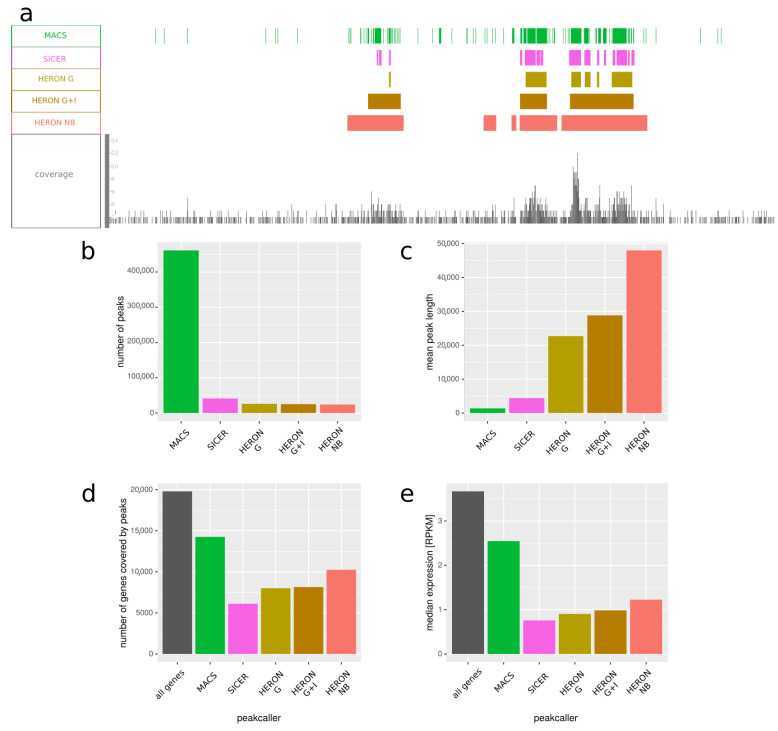
Comparison of peaks called by various methods on H3K27me3 ChIP-seq averaged over 10 samples. The data—including peaks called by MACS—is from Roadmap Epigenomics project. (**a**) Example representative genome fragment from fetal brain, showing peaks called by MACS, SICER, and HERON and the coverage of reads. HERON was run in three settings: “G”—with Gaussian distribution; “G + I”—with Gaussian distribution and with an input file; and “NB”—with negative binomial distribution. (**b**) Number of peaks called by the five approaches. (**c**) Mean length of peaks called by each peakcalling method. (**d**) Number of genes covered by peaks called by different peakcalling methods. First bar represents all the genes present in the annotation used. (**e**) Median expression in RPKM of all genes, compared to the expression of genes that overlap with peaks called by different peakcalling methods. We can see that MACS tends to call a lot of short peaks, which overlap more genes than peaks called by our approach. Furthermore, these genes have on average higher expression than the ones covered by the peaks called by HMM-based method, which might suggest many of them are not in fact within a H3K27me3 domain and the peaks that overlap with them are actually artifacts.

**Figure 5 ijms-22-08123-f005:**
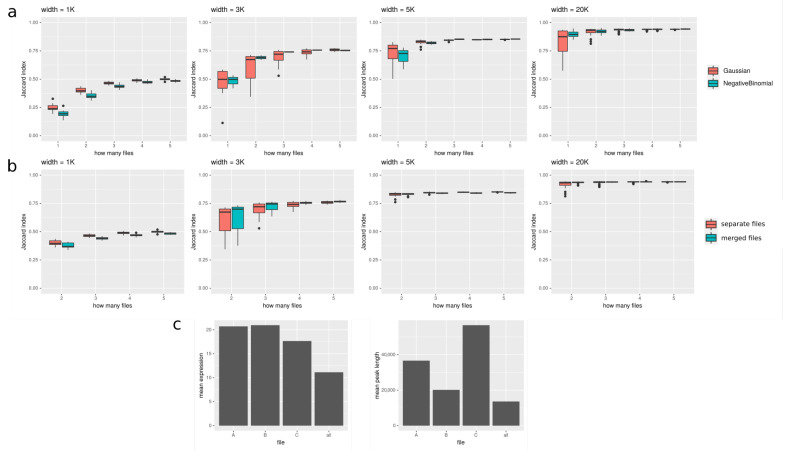
Peakcalling on multiple files; (**a**) and (**b**) are for simulated data, and (**c**) is for real data. (**a**) Plots show how Jaccard index (on *y* axis) between target and discovered peaks changes depending on number of input files (on *x* axis). It is shown for 4 various widths of target peaks: 1000, 3000, 5000, and 20,000. (**b**) Comparison of two approaches to using multiple files. One can see that, especially for longer peaks, there is no strong difference between them, while for peak width = 1000, treating files seperately yields better results, and for longer peaks, differences are smaller and sometimes favor merging all the used files. (**c**) Peakcalling on an example tissue from Roadmap Epigenomics (adult liver). Peaks called using all three available files are compared to three sets of peaks called using only single file. The left plot shows the mean expression of genes overlapping with called peaks, and the right one shows mean peak length of called peaks. Using all the files seems to give a conservative consensus peak set; the expression of genes overlapping with peaks called with this approach is substantially lower than the expression of genes overlapping with peaks called on single files, suggesting that using all the files yields more reliable results; at the same time, peaks called on all available files tend to be shorter than those called only on single one, what could result from the inevitable variability between the samples.

**Table 1 ijms-22-08123-t001:** Comparison of different peakcalling tools available with respect to their features including the newly proposed HERON method.

Software Feature	HERON	MACS	BayesPeak	SICER	PeakSeq
assumed signal	negative binomial	Poisson	negative	Poisson	binomial
distribution	Gaussian		binomial		
can handle replicates	yes	yes	no	no	yes
can use control signal	yes	yes	yes	yes	yes
input format	bam/sam/bedgraph	bam/sam/bed/eland	bed	bam/bed	sam/eland/tagAlign
species	any	any	any	set of available species	any
next window	no	yes	no	yes	yes
independence					
takes mappability	no	no	no	no	yes
into account					
scoring	posterior probability	*p*-value	posterior probability	*p*-value	*p*-value
method	coverage	*q*-value			*q*-value
		coverage

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
