# Peer review of "HERON: A Novel Tool Enables Identification of Long, Weakly Enriched Genomic Domains in ChIP-seq Data"

_ijms, 2021, doi:10.3390/ijms22158123_

Round 1
Reviewer 1 Report
The article by Anna Macioszek and Bartek Wilczynski entitled “A novel tool enables identification of long, weakly enriched genomic domains in ChIP-seq data” is a very good work that will be of interest for the scientific community. This manuscript can be published in the journal “International Journal of Molecular Sciences” after some minor revision.
My minor concerns are:
- The authors have to modify the title due to the fact that they have not shown that Neurofibromin1 expression level is lower in post-treatment sample than in pre-treatment samples. It is only a very light tendency in a small number of patients and the authors have the feeling it could be that acquired resistance could be related to NF1 downregulation during treatment.
- The authors have to add citations for “… epigenetic landscapes () or gene expression ()…” in line 14.
- The authors should double check the manuscript in regard of English language – some typos and grammar mistakes are present.
Author Response
The article by Anna Macioszek and Bartek Wilczynski entitled “A novel tool enables identification of long, weakly enriched genomic domains in ChIP-seq data” is a very good work that will be of interest for the scientific community. This manuscript can be published in the journal “International Journal of Molecular Sciences” after some minor revision.
We would like to thank the reviewer for a very quick response time.
My minor concerns are:
- The authors have to modify the title due to the fact that they have not shown that Neurofibromin1 expression level is lower in post-treatment sample than in pre-treatment samples. It is only a very light tendency in a small number of patients and the authors have the feeling it could be that acquired resistance could be related to NF1 downregulation during treatment.
This comment is likely a copy-paste error as our article is not dealing with Neurofibromin1 expression.
- The authors have to add citations for “… epigenetic landscapes () or gene expression ()…” in line 14.
We have added the required references.
- The authors should double check the manuscript in regard of English language – some typos and grammar mistakes are present.
We have carefully re-read the manuscript and improved the language
Reviewer 2 Report
In this work Macioszek et al. have developed a tool to identify the long weakly enriched genomic domains using ChIP-seq data. This study addresses a very important and long lasting problem of identifying these specific regions from ChIP-seq data. As such this work might be of interest to researchers in this field. However there are several major concerns that need to be addressed in order to strengthen the conclusions of this study:
1) Authors should compare their results with SICER which is one of the most well-known tools used for calling broad peaks from ChIP-seq data.
2) Throughout the text, authors talk about 'weakly enriched' peaks. Is there any specific definition for defining 'weakly enriched' peaks? may be with respect to some enrichment score?
3) Line 71 states that the use of HMM with continuous emissions returned good results. What do authors mean by good results? Is it on the basis of some statistical significance or the number of peaks reported? Please provide specifics.
4) It seems that the tool is based on HMM which has been used by several other tools previously for identifying ChIP-seq peaks. Please provide details on how is this tool different from the tools already known to be using HMM for peak-calling.
5) A schematic/workflow showing the steps used by this tool for peak calling must be provided. This would help readers understand the working principle of the tool easily.
6) Line 345: Is there any specific reason for choosing chr21 for the simulation study?
7) Authors must provide a descriptive table showing the features based on which this new tool is different than the existing peak calling tools.
8) Line 77 mentions the use of custom width. Is this custom width dependent on the kind of histone modification used for a study? How does the tool choose the custom width?
9) Does this tool work with the same specificity and sensitivity for the different histone modifications?
10) Line 84: what do authors mean by 'better results'. Please provide specifics.
11) According to line 120 window width is not crucial however the first step of the tool is dividing the genome into windows. Please justify.
12) Line 150: Authors mention that this tool returns false positive which is quite questionable. Please provide the specificity and sensitivity values for this tool. Also please justify why does this tool return more false positives than MACS.
13) Authors should also consider naming their tools. Use of the terms 'negative binomial' and 'gaussian' is confusing.
14) Is this tool species specific? Does the window width vary depending upon the species used for experimentation?
Author Response
In this work Macioszek et al. have developed a tool to identify the long weakly enriched genomic domains using ChIP-seq data. This study addresses a very important and long lasting problem of identifying these specific regions from ChIP-seq data. As such this work might be of interest to researchers in this field.
We would like to thank the reviewer for a quick and insightful review of our work.
However there are several major concerns that need to be addressed in order to strengthen the conclusions of this study:
1) Authors should compare their results with SICER which is one of the most well-known tools used for calling broad peaks from ChIP-seq data.
We have compared our results to SICER’s performance on both simulated and experimental data and added the analysis of these results to the figures and text. Indeed SICER’s performance on long peaks is better than MACS, however in the case of simulated data (Fig. 2) with long peaks (>5kbps) and low enrichment (enrichment<5) our tool is clearly outperforming SICER.
2) Throughout the text, authors talk about 'weakly enriched' peaks. Is there any specific definition for defining 'weakly enriched' peaks? may be with respect to some enrichment score?
Our results indicate that the lower the enrichment, the bigger the difference is between our peakcaller and other methods, however given the current results, we can say that the most appropriate situation for our tools are for the case of relative read enrichment (signal to noise ratio) of 2 to 5. We discuss it in the discussion section.
3) Line 71 states that the use of HMM with continuous emissions returned good results. What do authors mean by good results? Is it on the basis of some statistical significance or the number of peaks reported? Please provide specifics.
We have changed the language there to specify that by “good results” we mean both high (>.8) sensitivity and specificity
4) It seems that the tool is based on HMM which has been used by several other tools previously for identifying ChIP-seq peaks. Please provide details on how is this tool different from the tools already known to be using HMM for peak-calling.
Indeed, the idea of using HMMs for peak calling has been proposed in literature in different contexts. We have compared our results to the BayesPeak method which seems to be currently the most popular of these tools and provided more context in the introduction section.
5) A schematic/workflow showing the steps used by this tool for peak calling must be provided. This would help readers understand the working principle of the tool easily.
We have added a new figure (1) to the paper showing the schematic workflow of our method.
6) Line 345: Is there any specific reason for choosing chr21 for the simulation study?
Chromosome 21 is frequently used in similar contexts as it is one of the shorter chromosomes and contains chromatin features representative to all of the genome (which is not the case for X or Y chromosomes, for example)
7) Authors must provide a descriptive table showing the features based on which this new tool is different than the existing peak calling tools.
We have provided such a table (1) comparing several relevant tools for ChIP-seq peak-calling
8) Line 77 mentions the use of custom width. Is this custom width dependent on the kind of histone modification used for a study? How does the tool choose the custom width?
Most peak calling approaches contain a pre-processing step, where data is binned into windows of a fixed length. Our tool is no different and we have re-worded this fragment to not use the word custom, to make it clearer.
9) Does this tool work with the same specificity and sensitivity for the different histone modifications?
We have only tested it on H3K27me3, as it is the one that is most clearly matching the intended use of our tool (long, weakly enriched peaks). We expect that the performance of our tool on modifications with more sharp peaks (like H3K4me3 or H3K27ac) would be inferior to other tools MACS, based on the simulation studies. This is why we recommend usage of other tools in these scenarios.
10) Line 84: what do authors mean by 'better results'. Please provide specifics.
We have modified this sentence to clarify that we mean higher specificity peaks with lower expression of affected genes.
11) According to line 120 window width is not crucial however the first step of the tool is dividing the genome into windows. Please justify.
The sentence in question mentions “the choice of the window width is not crucial”, which might have been misleading. We have re-worded the sentence to clarify that indeed the method needs to make a choice of a window size, however the results quality is not particularly sensitive to this choice, i.e. we get good results for a wide range of window sizes
12) Line 150: Authors mention that this tool returns false positive which is quite questionable. Please provide the specificity and sensitivity values for this tool. Also please justify why does this tool return more false positives than MACS.
Our statement is based on the plot in Figure 3C (former figure 2C) that is showing that all methods have non-zero false discovery rate, which is clearly indicating some false positive predictions. Since these plots are from simulated data, we can be sure that these are indeed false positives. As we can see, the blue line corresponding to MACS is clearly lower than the other methods. We have also provided similar plots presenting specificity measures, however, as the specificity measures for all tools are very high (>98%), mostly due to the unbalanced nature of the peak calling problem (much more background genome, than peaks), we provide the sensitivity plots as supplementary figures and keep the FDR in the main text. Nonetheless, the overall findings remain the same, regardless of whether we use specificity or FDR to measure the amount of falsely positive predictions. In general, the presence of false positives is caused by the noise in the data leading to some localized enrichments that are usually not reproducible between replicates. This is why all methods improve in terms of false positives when presented with multiple replicates (as in Fig. 5). At the same time, in cases where HERON gives more false positives than MACS or SICER, it also gives many more true positives, which in our opinion is a good tradeoff.
13) Authors should also consider naming their tools. Use of the terms 'negative binomial' and 'gaussian' is confusing.
We have re-named our tool from peakcaller to HERON( HiddEn maRkov mOdel based peak fiNder) and modified the text and figures to make it clear
14) Is this tool species specific? Does the window width vary depending upon the species used for experimentation?
HERON is not species-specific. It can be run on any genome, provided that the sequence of the genome is known and the NGS reads can be mapped to it. We do not expect any particular dependence on the genome, other than the one we have mentioned before that the window size should not be larger than the expected domain size. However, given that our tool can detect large domains (>30kb) even with window size ~1000bp, we do not think that this needs to be changed for different genomes.

Round 2
Reviewer 2 Report
Authors have satisfactorily responded to the comments.